# The *GSTO2* (rs156697) Polymorphism Modifies Diabetic Nephropathy Risk

**DOI:** 10.3390/medicina59010164

**Published:** 2023-01-13

**Authors:** Dragana Pavlovic, Sinisa Ristic, Ljubica Djukanovic, Marija Matic, Marijana Kovacevic, Marija Pljesa-Ercegovac, Jovan Hadzi-Djokic, Ana Savic-Radojevic, Tatjana Djukic

**Affiliations:** 1Faculty of Medicine, University of East Sarajevo, 73300 Foca, Bosnia and Herzegovina; 2Academy of Medical Sciences of Serbian Medical Society, 11000 Belgrade, Serbia; 3Faculty of Medicine, University of Belgrade, 11000 Belgrade, Serbia; 4Institute of Medical and Clinical Biochemistry, 11000 Belgrade, Serbia; 5Serbian Academy of Sciences and Arts, 11000 Belgrade, Serbia

**Keywords:** type 2 diabetes mellitus, diabetic nephropathy risk, genetic polymorphisms, advanced glycation end products, glutathione transferase

## Abstract

*Background and Objectives*: In the development of type 2 diabetes mellitus (T2DM) and its complications, genetic and environmental factors play important roles. Diabetic nephropathy (DN), one of the major microangiopathic chronic diabetic complications, is associated with an increased risk of major cardiovascular events and all-cause mortality. The present study was designed to investigate the possible modifying effect of glutathione transferase polymorphisms (*GSTM1, GSTT1, GSTP1* rs1138272/rs1695, *GSTO1* rs4925 and *GSTO2* rs156697) in the susceptibility to T2DM and diabetic nephropathy. *Materials and Methods*: *GSTM1* and *GSTT1* deletion polymorphisms were determined by multiplex PCR, whereas *GSTO1, GSTO2*, and *GSTP1* polymorphisms were determined by the real-time PCR in 160 T2DM patients and 248 age- and gender-matched controls. Advanced glycation end products (AGEs) were measured by ELISA. *Results*: Among six investigated GST polymorphisms, a significant association between the *GST* genotypes and susceptibility for development of diabetes mellitus was found for the *GSTM1, GSTT1, GSTP1* (rs1138272) and *GSTO1* polymorphisms. When the *GST* genotypes’ distribution in diabetes patients was assessed in the subgroups with and without diabetic nephropathy, a significant association was found only for the *GSTO2* rs156697 polymorphism. Diabetic patients, carriers of the *GSTM1* null, *GSTT1* null and variant *GSTO1**AA genotypes, had significantly increased levels of AGEs in comparison with carriers of the *GSTM1* active, *GSTT1* active and referent *GSTO1**CC genotypes (*p* < 0.001, *p* < 0.001, *p* = 0.004, respectively). *Conclusions*: The present study supports the hypothesis that GST polymorphisms modulate the risk of diabetes and diabetic nephropathy and influence the AGEs concentration, suggesting the potential regulatory role of these enzymes in redox homeostasis disturbances.

## 1. Introduction

Diabetes mellitus (DM) represents one of the major public health problems worldwide, with continually increasing prevalence in the past few decades, especially in low- and middle-income countries [1]. As a result of deficiency in insulin action and/or secretion, this metabolic disorder characterized by chronic hyperglycemia leads to serious damage to many of the body’s systems over time. Moreover, DM is associated with an increased risk of major cardiovascular events and all-cause mortality [2]. Even with significant improvement in the clinical management of these patients, they develop a wide variety of fatal long-term macrovascular and microvascular complications [3]. As one of the most significant microvascular chronic complications, affecting almost 30–40% of diabetic patients, diabetic nephropathy (DN) represents the principal cause of chronic kidney disease (CKD) which leads to end-stage renal disease (ESRD), and almost half of the patients entering dialysis each year suffer from this condition [4,5].

The development of kidney injury in DM patients is clearly multifactorial, including various genetic and environmental factors [5]. Among them, the most recognized factor in DN etiopathogenesis is definitely long-lasting hyperglycemia, which seems to cause kidney damage by complex metabolic perturbations including increased activity of the polyol and the hexosamine pathways, as well as amplified intracellular formation of advanced glycation end products (AGEs) [6,7]. Moreover, prolonged hyperglycemia in diabetic patients induces increased reactive oxygen species (ROS) production, which consequently leads to oxidative stress (OS) [8,9]. There is a body of evidence that oxidative stress plays an important role in DM development [10,11,12], and it has been shown that polymorphism in antioxidant enzymes genes is associated with susceptibility to DM in various populations [13]. Moreover, association between several genetic variants related to antioxidant enzymes and type 2 DM has been confirmed by genome-wide association studies (GWAS), mostly in European descent population [14,15]. Thus, it could be proposed that genetic polymorphisms reducing the activity of antioxidant enzymes could increase a susceptibility to DN development as well [10,11].

Beyond well-characterized catalytic activity in the conjugation of reduced glutathione (GSH) to electrophilic centers on a wide range of substrates, glutathione transferases (*GST*s) exhibit various roles in redox homeostasis, which imply their importance in different oxidative stress-related diseases, including diabetes [16]. Moreover, functional gene polymorphisms in numerous isoforms within the cytosolic GST classes [12] may also affect their expression and activity. In particular, the association of the most extensively studied deletion (null) polymorphisms in *GST mu 1* (*GSTM1*) and *GST theta 1 (GSTT1*) genes, which results in a complete lack of corresponding enzymes [17], has been described to be involved in T2DM development and diabetes-related complications [18,19]. Moreover, the *GSTT1* null genotype or both the *GSTT1* and *GSTM1* null genotypes have been shown to be a genetic risk factor for the development of T2DM cardiovascular complications [20,21]. A meta-analysis suggests that there is a significantly increased risk of DN for the *GSTM1* null genotype, as well as the combination of null alleles for *GSTM1* and *GSTT1*. However, the authors found no correlation between DN and the individual *GSTT1* null genotype [22,23]. A possible clinical impact in disease susceptibility and response to oxidative stress [24] has also been shown for two single nucleotide polymorphisms (SNPs) of the *GST pi 1* (*GSTP1*) gene (rs1695 c.313A > G, p.IIe105Val and rs1138272 c.341C > T, p.Ala114Val). Regarding rs1695 polymorphism, the exchange of adenine for guanine at position 313 in codon 105 causes a change of amino acid isoleucine (Ile) for valine (Val), with consequently a lower activity of this isoform [24]. However, a recent meta-analysis found no association between the *GSTP1* Val allele rs1695 and T2DM susceptibility [23]. As a member of the GST class with distinctive catalytic activity in comparison to other GST classes, GST omega 1 (GSTO1) exhibits thiol transferase and deglutathionylase activity. Interestingly, a recent finding pointed out that reduced insulin content of the islet cells of diabetic Goto–Kakizaki (GK) rats was a consequence of *GSTO1* overexpression and its involvement in the regulation of a key insulin transcriptional factor, PDX1 [25]. In terms of the *GSTO1* SNP polymorphism (rs4925, c.419C > A, p.Ala140Asp), a change in its deglutathionylase activity has been shown [26], whereas in the case of the *GST omega 2* (*GSTO2*) polymorphism (rs156697, c.424A > G, p.Asn142Asp), a strong association between the *GSTO2**G variant allele and lower *GSTO2* gene expression was determined [27].

Hence, the present study was designed to investigate the possible modifying effect of *GST*s polymorphisms (*GSTM1*, *GSTT1*, *GSTP1*, *GSTO1* and *GSTO2*) in the susceptibility to T2DM and diabetic nephropathy as its major microvascular complication. Given the immense significance of AGEs in oxidative stress-related diabetes complications, the aim of this study was to investigate the relationship of *GST* polymorphisms with the plasma AGEs in patients with type 2 diabetes mellitus with and without diabetic nephropathy.

## 2. Materials and Methods

### 2.1. Study Design

Study group comprises 160 type 2 diabetes mellitus patients (89 men and 71 women, with an average age of 63.85 ± 10.09) recruited from the University Hospital of Foca (Foca, Bosnia and Herzegovina). All the patients were 18 years or older with a diagnosis of T2DM, without present macrovascular complications, malignant disease or acute infections. According to KDIGO guidelines, T2DM patients with chronic kidney disease (CKD) were classified based on persistent abnormalities of kidney structure and/or function for more than 3 months, an elevated urine albumin/creatinine ratio (UACR) (30 mg/g [3 mg/mmol]) and a reduced estimated glomerular filtration rate, eGFR (eGFR < 60 mL/min per 1.73 m^2^) [28]. The equation used was Modification of Diet in Renal Disease (MDRD) GFR equation that estimates glomerular filtration rate based on creatinine and patient characteristics. If T2DM patients with CKD had hematuria and/or pyuria, they were excluded from the study.

Age- and gender-matched healthy control group included 248 individuals (133 men, 115 women; average age 62.99 ± 10.37) whose DNA and genotyping results were taken from the biobank of the Institute of Biochemistry of Faculty of Medicine University in Belgrade. The “Declaration of Helsinki” and national and international ethical guidelines were followed during this study with approval obtained from the Ethics Committee of the Faculty of Medicine in Foca (approval number: 01-2-1, 1 November 2016). All recruited subjects were given an informed written consent to participate in the study.

For each study participant, demographic and clinical data were obtained together with anthropometric (weight, height and body mass index) and vital sign (blood pressure, pulse rate and temperature) measurements. A 3 mL of peripheral blood was collected in EDTA vacutainers. Urine samples were collected in urine containers and centrifuged at 4000 rpm for 5 min. The supernatant was stored in labeled Eppendorf tubes at −70 °C until further analysis of creatinine and protein in urine. Creatinine and protein were estimated using commercially available kits (Human, Wiesbaden, Germany) on HUMA–STAR 600 automated clinical chemistry analyzer in each urine sample. UPCR (urine protein to creatinine ratio) was calculated to determine renal status./VI-7).

### 2.2. DNA Isolation and Glutathione Transferases Genotyping

A total DNA was purified from EDTA-anticoagulated peripheral blood obtained from the study participants using PureLink™ Genomic DNA Mini Kit (Thermo Fisher Scientific, Carlsbad, CA, USA). Multiplex PCR was used for detection of the *GSTM1* and *GSTT1* gene deletion polymorphisms in accordance with the method by Abdel-Rahman et al. [29]. For each of the two genes, forward (Pf) and reverse (Pr) primers were used. Additionally, primers for *CYP1A1* gene were added to the mix as internal control. PCR conditions were as follows: initial denaturation at 94 °C for 4 min, followed by 30 cycles of short denaturation, annealing of primers and extension (30 s at 94 °C, 30 s at 59 °C, and 45 s at 72°, respectively), with the final 5 min extension at 72 °C. The PCR reaction was performed by Mastercycler pro S (Eppendorf, Hamburg, Germany). Visualization of the gene fragments amplified by PCR reaction was performed on a 2% agarose gel using SYBR Safe DNA gel stain (Thermo Fisher Scientific). The method enabled detection of only the presence or absence of the gene, marked as active when at least one allele was present (homozygote or heterozygote) and null with complete deletion of both alleles (homozygote). The presence of the GSTM1-active genotype was detected by the band at 215 bp, while the band at 480bp indicated the presence of GSTT1-active genotype.

*GSTO1* rs4925, *GSTO2* rs156697, *GSTP1* rs1695 and *GSTP1* rs1138272 polymorphisms were determined by the real-time PCR on Mastercycler ep realplex (Eppendorf, Hamburg, Germany), using TaqMan Drug Metabolism Genotyping assays (Life Technologies, Applied Biosystems, Carlsbad, CA, USA). Assays’ IDs were as follows: C_3223136_1, C_3223136_1, C_3237198_20 and C_1049615_20, respectively.

### 2.3. Advanced Glycation End Product (AGE) Measurement

Plasma AGEs were measured in EDTA samples obtained from fasting venous blood, which were stored at −80 °C until analysis. Plasma Advanced Glycation End Products (AGEs) levels were measured by an AGEs competitive ELISA Kit (OxiSelect™, Cell Biolabs, San Diego, CA, USA), which used glyceraldehyde-derived AGE-BSA as a standard. The principle is based on determination of concentration of AGE adduct in protein samples by comparing its absorbance with that of a known AGE-BSA standard curve. First, an AGE conjugate was coated on an ELISA plate. The unknown AGE protein samples or AGE-BSA standards were then added to the AGE conjugate preabsorbed on plate. After a brief incubation, an anti-AGE polyclonal antibody (1:1000) was added, followed by an HRP conjugated secondary antibody (1:1000). The enzyme reaction was stopped by adding stop solution to each well. The absorbances were read on microplate reader using 450 nm as the primary wave length. The content of AGE protein adducts in unknown samples was determined by comparison with a predetermined AGE-BSA standard curve.

### 2.4. Statistical Analysis

Statistical analysis was performed in SPSS software version 17.0 (SPSS Inc., Chicago, IL, USA). Selected characteristics of control group and diabetic patients were compared. Differences between categorical variables, as well as Hardy–Weinberg equilibrium for respective genotypes were tested using χ2-test, whereas, depending on data distribution, Student’s t test or Mann–Whitney test were used for continuous variables. The effect of genetic variations on the risk of diabetes and nephropathy development was computed by odds ratio (OR) and 95% confidence interval (CI) with the use of logistic regression. Level of statistical significance was set at *p* < 0.05.

## 3. Results

### Demographic and Clinical Characteristics

A total number of 160 diabetes mellitus (T2DM) patients and 248 age- and gender-matched controls were included in the study. Baseline characteristics of patients and controls are summarized in Table 1.

As presented, a statistically significant difference was observed among T2DM patients and controls regarding body mass index (BMI) and hypertension. Namely, almost 77% of patients had hypertension in comparison with 37% of individuals with hypertension in the control group.

Patients with diabetes were further divided into two subgroups, with one comprising 91 patients without diabetic nephropathy and the other comprising 69 patients with diabetic nephropathy. No statistically significant difference regarding age, gender, BMI and glucose concertation was observed between the subjects divided into these subgroups, but a statistically significant difference was observed regarding hypertension, duration of diabetes, hemoglobin concentration (Hb), urea, creatinine, estimated glomerular filtration rate (eGFR), urine protein and urine protein/creatinine ratio (UPCR) between these two groups of diabetic patients (Table 2).

The distribution of specific genotypes among diabetic patients and healthy controls is presented in Table 3. Among six investigated *GST* polymorphisms, a significant association between GST genotype and susceptibility for development of diabetes mellitus was found for the *GSTM1, GSTT1, GSTP1* (rs1138272) and *GSTO1* (rs4925) polymorphisms. Individuals with the GSTM1 null genotype were more prone to develop diabetes in comparison with *GSTM1* active carriers (OR = 1.97, 95%CI = 1.14–3.40, *p* = 0.015). On the other hand, carriers of the *GSTT1* null genotype were more protected compared to *GSTT1* active carriers (OR = 0.31, 95%CI = 0.18–0.56, *p* < 0.001). Namely, carriers of the heterozygous *GSTP1**CT rs1138272 genotype were in 3.4-fold higher odds compared to the carriers of the *GSTP1**CC genotype to develop diabetes (OR = 3.43, 95%CI = 1.53–7.70, *p* = 0.003). On the contrary, individuals with the *GSTO1**CA genotype had significantly lower odds of symptomatic diabetes development compared to the carriers of the wild-type *GSTO1**CC genotype (OR = 0.28, 95%CI = 0.15–0.51, *p* < 0.001).

When *GST* genotypes’ distribution in diabetes patients and controls was assessed in the subgroups with and without diabetic nephropathy, a significant association was found only for *GSTO2* rs156697 polymorphisms (Table 4). Namely, we observed that individuals with the *GSTO2**AG genotype were 2.6-fold more prone for symptomatic diabetic nephropathy development (OR = 2.59, 95%CI = 1.11–6.05, *p* = 0.028) in comparison to the carriers of the wild-type *GSTO2**AA genotype. Similarly, individuals with the *GSTO1**AA genotype rs4925 had higher odds of diabetic nephropathy development compared to the carriers of the wild-type *GSTO1**CC genotype, however with borderline significance (OR = 3.81, 95%CI = 0.85–17.09, *p* = 0.081).

Advanced Glycation End Products (AGEs) concentration with regard to specific genotype in diabetic patients is presented on Figure 1. Patients who were carriers of the *GSTM1* null, *GSTT1* null and variant *GSTO1**AA genotypes had significantly increased levels of AGEs in comparison with carriers of the *GSTM1* active, *GSTT1* active and referent *GSTO1**CC genotypes (*p* < 0.001, *p* < 0.001, *p* = 0.004, respectively).

Advanced Glycation End Products (AGEs) concentration with regard to specific genotype in diabetic patients without nephropathy is presented in Figure 2. The patients who were carriers of the *GSTM1* null and *GSTT1* null genotypes had significantly increased levels of AGEs in comparison with carriers of the *GSTM1* active and *GSTT1* active genotypes (*p* < 0.001, *p* < 0.010, respectively). The carriers of the *GSTP1**GG genotype had significantly decreased AGEs concentration compared to *GSTP1**AG carriers (*p* = 0.025).

Advanced Glycation End Products (AGEs) concentration with regard to specific genotype in diabetic patients with nephropathy is presented on Figure 3. The patients who were carriers of *GSTM1* null, *GSTT1* null and variant *GSTO1**AA genotypes had significantly increased levels of AGEs in comparison with carriers of the *GSTM1* active, *GSTT1* active and referent *GSTO1**CC genotypes (*p* < 0.001, *p* = 0.036, *p* = 0.053, respectively).

## 4. Discussion

Among six glutathione transferase (*GST*) polymorphisms investigated in this study, a significant association between the *GSTM1*, *GSTT1*, *GSTP1* (rs1138272) and *GSTO1* polymorphisms and T2DM development was found. Regarding susceptibility to DN development, only carriers of at least one variant *GSTO2* allele were at increased risk. Moreover, diabetic patients, carriers of the *GSTM1* null, *GSTT1* null and variant *GSTO1**AA genotypes had significantly increased levels of advanced glycation end products in comparison with carriers of the *GSTM1* active, *GSTT1* active and referent *GSTO1**CC genotypes.

It has been well known that the major cause of increased reactive oxygen species (ROS) generation in T2DM might be related to the impairment of the mitochondrial electron transport chain, as well as increased enzymatic activities of NADPH oxidase and xanthine oxidase [30]. Pancreatic β-cells have emerged as a recognized target of oxidative stress-induced damage, which partially explains the progressive deterioration of the β–cell function in T2DM [31]. Enhanced mitochondrial respiration induced by fuels oversupply with a consequent increase in ROS production, simultaneously with damage of β -cells due to hypoxia, represents the main underlying mechanism of disrupted redox homeostasis associated with pancreatic beta-cell dysfunction. Despite high ROS production as a result of overstimulation of beta cells with glucose and other fuels, the expression of antioxidant defense genes is unusually low (or disallowed) in those cells. Several genome–wide association studies demonstrated an effect of T2DM risk-associated gene variants on beta-cell function rather than on insulin sensitivity [9].

As an enzyme superfamily exhibiting primarily detoxifying activity towards a wide range of electrophilic substrates and inactivation of secondary metabolites formed during oxidative stress [32], polymorphisms in genes encoding different classes of glutathione transferases (*GST*s) could affect activity of corresponding enzymes and would make the cell more susceptible to disturbances in redox homeostasis [11]. To our knowledge, this is the first report on the association of the GSTP1 (rs1138272) polymorphism with a risk of T2DM development. Given the important pleiotropic redox functions of GSTP1, it seems reasonable to suggest its risk-associated modifying role. So far, the data of several meta-analysis, which have assessed the association of *GSTM1* and *GSTT1* deletion polymorphisms with T2DM and its complications, are inconsistent [23,33,34]. Thus, Yi et al. [33] and Zhang et al. [34] found significant associations between the GSTM1 null or *GSTT1* null genotype and diabetes risk. Considering the effect of racial background, several ethnic population-based studies have been designed to evaluate the association between the above *GST* deletion polymorphisms and T2DM. Thus, it has been shown that the GSTT1 null genotype was associated with T2DM in Chinese and Egyptian populations, while a relation of the *GSTM1* null genotype with T2DM risk was found among Asians but not Caucasians. Hovnik et al. [35] concluded that carriers of both the *GSTM1* null and *GSTT1* null genotypes, and a consequent complete lack of GSTM1 and GSTT1 enzyme activity, were not at higher risk of developing macroangiopathic complications in type 1 diabetes. The results of our study indicate an approximately 2-fold higher risk of developing T2DM in carriers of the *GSTM1* null genotype, while a protective effect has been shown for carriers of the *GSTT1* null genotype. Further, our results on higher plasma concentrations of AGEs in T2DM patients, carriers of either the *GSTM1* null or *GSTT1* null genotype, suggest that deficient activity of these enzymes would make individuals more susceptible to the oxidative stress-related complications of disease.

It seems that involvement of metabolic disturbances affecting several signaling pathways is crucial for diabetes kidney disease development. Free fatty acids (FFA) are increased in subjects with T2DM, and they have been shown to induce inflammatory cytokine production via Toll-like receptor-TLR4 signaling [36]. Furthermore, hyperglycemia also induces overproduction of advanced glycation end products (AGEs) as the result of glyoxal oxidation, 3-deoxyglucoson formation and fragmentation of glyceraldehyde 3-phosphate into methylglyoxal. Modified plasma proteins by AGEs allow their binding to cell surface receptors, such as the receptor for AGEs (RAGE), or macrophage scavenger receptors. Highly abundant AGEs in the diabetic milieu of the kidneys increase RAGE expression [37]. RAGE activation by AGEs leads to the ROS generation and amplifies inflammation that aids in the establishment of a chronic inflammatory state in the kidneys culminating in gradual loss of kidney architecture and function [38]. In addition to reducing antioxidant enzymes and cellular glutathione levels, RAGE activation results in the up-regulation of NADPH oxidase, nitric oxide synthase (NOS), and cyclooxygenase (COX), the events that exacerbate and intensify the inflammation [39].

Another GST investigated in this study, glutathione transferase omega 1 (GSTO1), is an atypical GST with the important role in proinflammatory signaling [39,40,41]. It was shown that GSTO1-1 can modulate the proinflammatory lipopolysaccharide (LPS)/Toll-like receptor (TLR-4)-induced activation of the nuclear factor kappa B (NF-κB) pathway in macrophages [42]. Additionally, Hughes et al. examined a role for GSTO1-1 in NLRP3 inflammasome activation and found that GSTO1-1 promotes NLRP3 activation [41]. Furthermore, Menon et al. have noticed the decrease in IL-1β expression in GSTO1-1 deficient mice treated with LPS, providing us with a possible mechanism by which GSTO1-1 deficiency attenuates insulin resistance in mice on a high fat diet [40]. This finding suggested the possibility that GSTO1-1 inhibitors may be of value in the treatment of T2DM since IL-1β inhibitors have already been found to be effective in modulating insulin resistance [43,44]. What we know so far is that *GSTO1* rs4925 polymorphism would cause changes mainly in deglutathionylase activity of this GST enzyme [26,45,46]. Namely, Menon and Board have shown that the *GSTO1**A variant allele has lower deglutathionylase activity and higher activity in the forward glutathionylation reaction, opposite of the *GSTO1**C wild-type allele [26]. What was found when the risk for diabetes or diabetes nephropathy was assessed in our study was that carriers of at least one *GSTO1* rs4925 variant allele (*CA+ AA) were less prone to T2DM development. Since the wild-type allele with its deglutathionylase activity is responsible for the pro-inflammatory effects of GSTO1, we might speculate that carriers of at least one variant allele are in that way more protected from inflammation, since inflammation is a common feature in subjects with T2DM. On the other hand, when AGEs concentrations were examined, a statistically significant increase of AGEs concentration was found in carriers of the *GSTO1* variant genotype (*GSTO1**AA) compared to the AGEs concentration found in the *GSTO1* wild-type genotype (*GSTO1**CC). Similar findings were encountered in diabetic patients with and without diabetic nephropathy. Although we expected the AGEs concentration to be higher in carriers of the “pro-inflammatory” *GSTO1**CC genotype, an interesting point was observed when we searched for the possible explanation. The concentration of AGEs in patients without diabetic nephropathy carriers of the *GSTO1**AA genotype was 27.36 µg/mL compared to the concentration of 38.42 µg/mL in patients with diabetic nephropathy carriers of the same genotype. Diabetic nephropathy (DN) increases morbidity and mortality among people living with diabetes [47]. The aforementioned result gave us the opportunity to speculate that the *GSTO1**AA “anti-inflammatory” genotype protected the patients from developing diabetic nephropathy, the long-time existing complication of type 2 diabetes mellitus.

A reaction that is atypical for other *GST*s but catalyzed by both GSTO1 and GSTO2 is dehydroascorbate reductase (DHAR) activity, a reaction important for ascorbic acid level preservation. GSTO2 is suggested to have the highest DHAR activity in the cells, which is up to 100 times higher than GSTO1 DHAR activity [48]. Only a few studies have investigated *GSTO2* polymorphisms in non-malignant diseases. Cimbaljevic et al. reported that individuals with the variant *GSTO2**GG genotype were at higher risk of ESRD development compared to carriers of the *GSTO2**AA genotype [49]. In our study, we found that individuals with the *GSTO2**AG genotype were more prone to symptomatic diabetic nephropathy development in comparison to the carriers of the wild-type *GSTO2**AA genotype. The patients who were carriers of the *GSTO2**AG genotype had lower AGEs concentration in comparison with the *GSTO2**AA genotype carriers, but the difference was not significant.

Certain limitations should be considered in our study. A relatively small number of the study participants might be the source of potential biases that may have influenced the study findings. The study participants were Caucasians only; therefore, the possible effect of ethnicity could not be assessed. Future studies with rigorous designs are needed to confirm the findings of our research.

## 5. Conclusions

Despite the above-mentioned limitations, the present study supports the hypothesis that GST modulates the risk of diabetes and diabetic nephropathy and influences the AGEs concentration. Given the large number of genes that have been uncovered, our understanding of their role in causing diabetes and diabetic nephropathy is a considerable challenge. DN is a heterogeneous disease associated with an extremely high risk of cardiovascular events, chronic kidney disease progression and disability. By improving our knowledge of the diabetes and diabetic nephropathy genetics and the association between individual phenotype and genotype data, as well as the underlying mechanisms, we will be in the position to achieve individual medicine practices and develop target therapeutics.

## Figures and Tables

**Figure 1 medicina-59-00164-f001:**
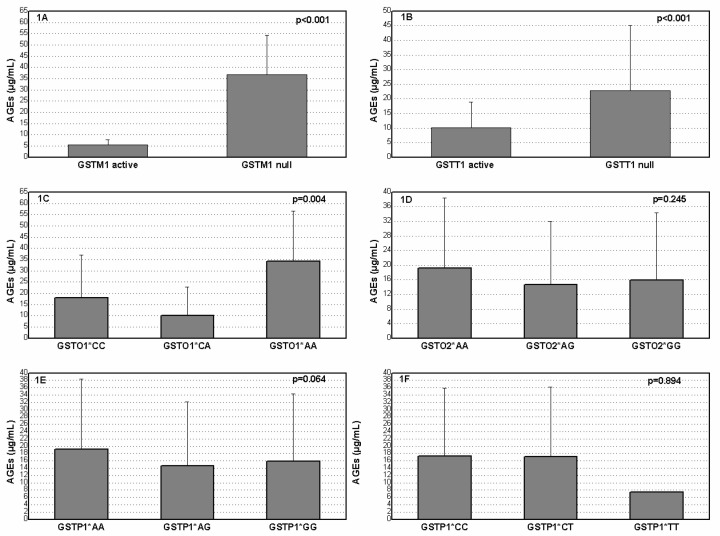
Plasma AGEs level in diabetes mellitus patients according to *GSTM1* (**A**), *GSTT1* (**B**), *GSTO1* rs4925 (**C**), *GSTO2* rs156697 (**D**), *GSTP1* rs1695 (**E**) and *GSTP1* rs1138272 (**F**) polymorphisms.

**Figure 2 medicina-59-00164-f002:**
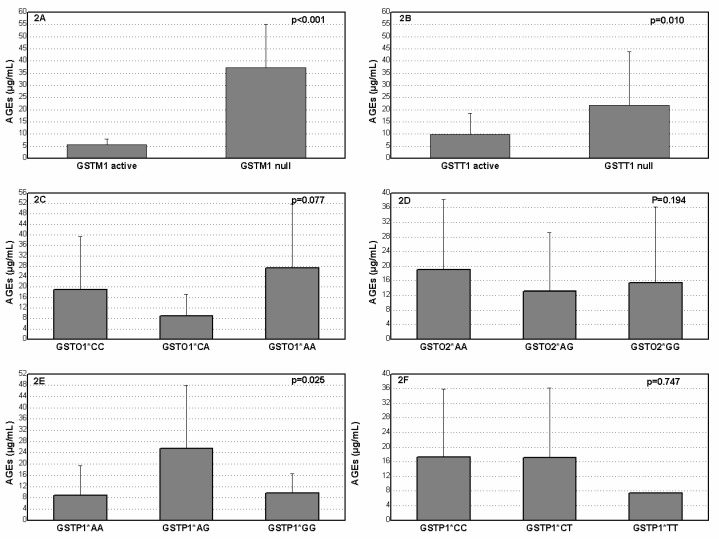
Plasma AGEs level in diabetes mellitus patients without nephropathy according to *GSTM1* (**A**), *GSTT1* (**B**), *GSTO1* rs4925 (**C**), *GSTO2* rs156697 (**D**), *GSTP1* rs1695 (**E**) and *GSTP1* rs1138272 (**F**) polymorphisms.

**Figure 3 medicina-59-00164-f003:**
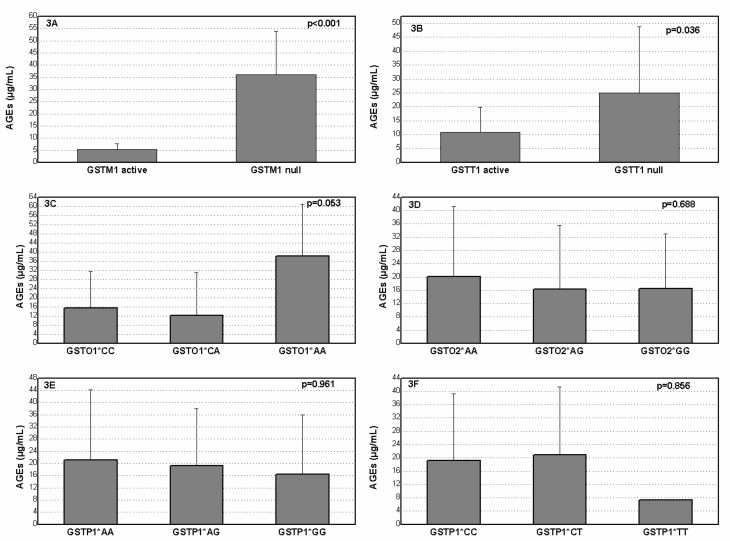
Plasma AGEs level in diabetes mellitus patients with nephropathy according to *GSTM1* (**A**), *GSTT1* (**B**), *GSTO1* rs4925 (**C**), *GSTO2* rs156697 (**D**), *GSTP1* rs1695 (**E**) and *GSTP1* rs1138272 (**F**) polymorphisms.

**Table 1 medicina-59-00164-t001:** Baseline characteristic of type 2 diabetes mellitus (T2DM) patients and age and gender matched controls.

	T2DM Patients n = 160	Controls n = 248	*p*
Age (years) ^a^	63.85 ± 10.09	62.99 ± 10.37	0.171
Gender, n (%) ^b^			
Male	89 (55.6)	133 (53.6)	
Female	71 (44.4)	115 (46.4)	0.693
Hypertension, n (%) ^b^			
No	37 (23.1)	157 (63.3)	
Yes	123 (76.9)	91 (36.7)	<0.001
BMI (kg/m^2^) ^a^	28.29 ± 4.36	26.26 ± 4.49	<0.001

^a^ Mean ± SD; ^b^ percentage; n, number of samples.

**Table 2 medicina-59-00164-t002:** Demographic and clinical characteristics of type 2 diabetes mellitus (T2DM) with and without diabetic nephropathy (DN).

	T2DM Patients		
	without DN (n = 91)	with DN (n = 69)	*p*
Age (years) ^a^	65.04 ± 9.13	62.28 ± 11.10	0.086
Gender, n (%) ^b^			
Male	49 (53.8)	40 (58.0)	
Female	42 (46.2)	29 (42.0)	0.603
Hypertension, n (%) ^b^			
No	12 (13.2)	25 (36.2)	
Yes	79 (86.8)	44 (63.8)	<0.001
BMI (kg/m^2^) ^a^	28.33 ± 4.18	28.23 ± 4.66	0.900
Duration of T2DM (years)	9.79 ± 7.54	12.82 ± 7.84	0.023
Glucose (mmol/L)	11.40 ± 4.46	10.99 ± 4.55	0.616
Hb (g/L)	140.00 ± 16.23	133.75 ± 15.22	0.030
HbA1c (%)	8.19 ± 1.68	8.84 ± 1.82	0.016
Urea(mmol/L) ^c^	5.80 (4.47–6.92)	8.60 (5.60–20.4)	<0.001
Creatinine(µmol/L) ^c^	73.60 (64.1–84)	103 (68.65–722)	<0.001
eGFR (ml/min/1.73m^2^) ^c^	95 (85–106.3)	82 (57.6–97.9)	0.001
Urine protein (mg/L) ^c^	0.11 (0.08–0.14)	0.42 (0.20–0.82)	<0.001
Urine protein/creatinine ratio (mg/mmol) ^c^	14.00 (9.00–31.60)	49.50 (22.60–97.30)	<0.001
AGEs (µg/mL)	16.66 ± 18.50	18.19 ± 19.20	0.711

^a^ Mean ± SD; ^b^ Percentage; ^c^ Median (min-max); n, number of patients.

**Table 3 medicina-59-00164-t003:** GST genotypes in relation to the risk of diabetes mellitus development.

*GST* Genotype	T2DM Patients n, %	Controls n, %	OR (95%CI) ^a^	*p* ^b^
*GSTM1*				
Active	86 (62.3)	119 (50.0)	1.00 ^c^	0.015
Null	52 (37.7)	119 (50.0)	1.97 (1.14–3.40)	
*GSTT1*				
Active	68 (49.3)	189 (79.4)	1.00 ^c^	
Null	70 (50.7)	49 (20.6)	0.31 (0.18–0.56)	<0.001
*GSTP1* rs1695				
AA	64 (41.6)	110 (44.5)	1.00 ^c^	
AG	63 (40.9)	99 (40.1)	1.02 (0.57–1.82)	0.945
GG	27 (17.5)	38 (15.4)	1.20 (0.55–2.64)	0.647
AG + GG	90 (58.4)	137 (55.5)	1.07 (0.62–1.82)	0.810
*GSTP1* rs1138272				
CC	114 (74.0)	196 (89.5)	1.00 ^c^	
CT	38 (24.7)	22 (10.0)	3.43 (1.53–7.70)	0.003
TT	2 (1.3)	1 (0.5)	NA	NA
CT + TT	40 (26.0)	23 (10.5)	3.53 (1.58–7.88)	0.002
*GSTO1* rs4925				
CC	94 (62.3)	85 (38.3)	1.00 ^c^	
CA	45 (29.8)	114 (51.4)	0.28 (0.15–0.51)	<0.001
AA	12 (7.9)	23 (10.4)	0.51 (0.18–1.46)	0.208
CA + AA	57 (37.7)	137 (61.7)	0.31 (0.17–0.55)	<0.001
*GSTO2* rs156697				
AA	64 (47.4)	93 (42.7)	1.00 ^c^	
AG	47 (34.8)	104 (47.7)	0.77 (0.43–1.39)	0.391
GG	24 (17.8)	21 (9.6)	1.79 (0.72–4.45)	0.208
AG + GG	71 (52.6)	125 (57.3)	0.93 (0.53–1.62)	0.803

^a^ OR, odds ratio, CI, confidence interval; ^b^ value and OR are obtained from logistic regression model adjusted for age, gender, body mass index, hypertension; ^c^ reference group; n, number of samples.

**Table 4 medicina-59-00164-t004:** GST genotypes in relation to the risk of diabetes mellitus nephropathy development.

	T2DM Patients			
*GST* Genotype	without DN n, %	with DN n, %	OR (95%CI) ^a^	*p* ^b^
*GSTM1*				
Active	57 (64.0)	29 (59.2)	1.00	
Null	32 (36.0)	20 (40.8)	1.25 (0.58–2.66)	0.566
*GSTT1*				
Active	41 (46.1)	27 (55.1)	1.00	
Null	48 (53.9)	22 (44.9)	1.00 (0.96–1.05)	0.904
*GSTP1* rs1695				
AA	35 (39.3)	29 (44.6)	1.00 ^c^	
AG	36 (40.4)	27 (41.5)	0.70 (0.30–1.62)	0.404
GG	18 (20.2)	9 (13.8)	0.88 (0.30–2.59)	0.816
AG + GG	54 (60.7)	36 (55.4)	0.75 (0.34–1.62)	0.463
*GSTP1* rs1138272				
CC	66 (74.2)	48 (73.8)	1.00 ^c^	
CT	22 (24.7)	16 (24.6)	0.61 (0.23–1.62)	0.318
TT	1 (1.1)	1 (1.5)	/	/
CT + TT	23 (25.8)	17 (26.2)	0.70 (0.27–1.80)	0.455
*GSTO1* rs4925				
CC	58 (69.9)	36 (52.9)	1.00 ^c^	
CA	21 (25.3)	24 (35.3)	1.47 (0.60–3.64)	0.400
AA	4 (4.8)	8 (11.8)	3.81 (0.85–17.09)	0.081
CA + AA	25 (30.1)	32 (47.1)	1.83 (0.80–4.18)	0.150
*GSTO2* rs156697				
AA	48 (55.2)	16 (33.3)	1.00 ^c^	
AG	25 (28.7)	22 (45.8)	2.59 (1.11–6.05)	0.028
GG	14 (16.1)	10 (20.8)	2.25 (0.77–6.57)	0.140
AG + GG	39 (44.8)	32 (66.7)	2.48 (1.14–5.40)	0.022

^a^ OR, odds ratio, CI, confidence interval; ^b^ value and OR are obtained from logistic regression model adjusted for age, gender, body mass index, hypertension; ^c^ reference group; n, number of samples.

## Data Availability

The data supporting the reported results can be provided upon request in the form of datasets available at the Clinic of Urology, University Clinical Centre of Serbia and the Institute of Medical and Clinical Biochemistry, Faculty of Medicine, University of Belgrade.

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
