# Peer review of "The GSTO2 (rs156697) Polymorphism Modifies Diabetic Nephropathy Risk"

_medicina, 2023, doi:10.3390/medicina59010164_

Round 1
Reviewer 1 Report
It is a well conducted study and a well written manuscript. My compliments to the authors for the same. However, there are some concerns/suggestions can help to improve the article.
1) In table 2, please put the sample size under each subgroup. N=91 under T2DM without DN subgroup and n= 69 under T2DM with DN subgroup.
2) Also in table 2, Put the units of glucose, Hb and HbA1c.
3) How was GFR evaluated? By clearance method or estimated by an equation? If estimated, please refer it as eGFR and name the used equation in the methods section.
4) Table 3 needs to be revised. Its title is “Combined effect of smoking and GPX1 and SOD2 genotypes on the risk of UBC” which is incompatible with its content as well as the overall study.
5) Also in table 3, the total number of T2DM patients with DN was 77 while the actual number in subjects and methods section for this subgroup was 69!!!
6) The authors mentioned in table 2 that minimum and maximum values of GFR in T2DM with DN were 57.5 and 97.9 while in table 3 there were 27 patients from the same subgroup had GFR less than 15. Did this subgroup contain patients with renal failure?
7) Explain the different numbers of patients and controls in genotyping analysis in table 4 than the actual numbers of individuals involved in the study? 138 patients from 160 were analyzed for GSTM1 ang GSTT1 genotypes and 154 patients were analyzed for GSTP1 genotyping, 151 for GSTO1, 135 for GSTO2 and also for controls. Why Not all the patients and controls had been evaluated in genotyping?
8) Type 2 diabetes mellitus was abbreviated in lines 175 and 178 in page 4 as DM2T. Please revise the manuscript and change all DM2T to T2DM.
Overall, It is a good effort by the authors
Author Response
1) In table 2, please put the sample size under each subgroup. N=91 under T2DM without DN subgroup and n= 69 under T2DM with DN subgroup.
2) Also in table 2, Put the units of glucose, Hb and HbA1c.
We thank the reviewer for all the comments. The overmentioned units of glucose, Hb and HbA1c and the number of sample size were added in Table 2.
3) How was GFR evaluated? By clearance method or estimated by an equation? If estimated, please refer it as eGFR and name the used equation in the methods section.
The equation used was Modification of Diet in Renal Disease (MDRD) GFR equation that estimates glomerular filtration rate based on creatinine and patient characteristics. We thank the reviewer for the comment and we added the explanation in the Materials and Methods section.
4) Table 3 needs to be revised. Its title is “Combined effect of smoking and GPX1 and SOD2 genotypes on the risk of UBC” which is incompatible with its content as well as the overall study.
We thank the reviewer for the comment. The title of the Table 3 was changed to “Patients with diabetic nephropathy classified into stadiums according to GFR” but the table was deleted later (the explanation is given below).
5) Also in table 3, the total number of T2DM patients with DN was 77 while the actual number in subjects and methods section for this subgroup was 69!!!
We thank the reviewer for the comment. This table was prepared for the initial versions of the paper and displayed in this paper by mistake. Namely, the authors have decided to present in this paper the results of 69 patients with diabetic nephropathy and normal kidney function or a slight decrease in eGFR. After careful reading of the manuscript, the authors have decided the table 3 is redundant and the table has been deleted.
6) The authors mentioned in table 2 that minimum and maximum values of GFR in T2DM with DN were 57.5 and 97.9 while in table 3 there were 27 patients from the same subgroup had GFR less than 15. Did this subgroup contain patients with renal failure?
We thank the reviewer for his careful observation. In this paper, the authors have presented the results of 69 patients with diabetic nephropathy and normal kidney function or a slight decrease in eGFR. The table 3 was prepared for the initial versions of the paper and displayed in this paper by mistake, hence we deleted it but the numbers for eGFR in table 2 are correct.
7) Explain the different numbers of patients and controls in genotyping analysis in table 4 than the actual numbers of individuals involved in the study? 138 patients from 160 were analyzed for GSTM1 ang GSTT1 genotypes and 154 patients were analyzed for GSTP1 genotyping, 151 for GSTO1, 135 for GSTO2 and also for controls. Why Not all the patients and controls had been evaluated in genotyping?
We thank the reviewer for the comment. All the patients and controls had been evaluated in genotyping. Unfortunately, not all genotyping was successful. The most frequent reason for failure to provide a reaction product was the purity of the sample (assessed by the UV spectrometry) which interfered with amplification reactions.
8) Type 2 diabetes mellitus was abbreviated in lines 175 and 178 in page 4 as DM2T. Please revise the manuscript and change all DM2T to T2DM.
We thank the reviewer for the made observation. We changed DM2T to T2DM as suggested.
Reviewer 2 Report
The study highlights significant findings in terms of significant association between risk of diabetic nephropathy and GSTO2 polymorphism. The study is well executed however, the following corrections are required
1. The sample size is small in relation to the study.
2. The authors can include a flow chart to highlight the inclusion and exclusion criteria of patients and the studies performed
3. The discussion is hard to follow. The authors should specifically discuss their results in comparison with the previous findings
Author Response
- The sample size is small in relation to the study.
We agree with the reviewer that the sample size is relatively small and we mentioned that as one of the limitations of the study (last paragraph in the Discussion section) but the power of the study, as calculated by a statistician, was adequate to draw the conclusions.
- The authors can include a flow chart to highlight the inclusion and exclusion criteria of patients and the studies performed
We thank the reviewer for the suggestion. We have tried to explain more clearly the inclusion and exclusion criteria and the analysis performed in the Methods and Materials section.
- The discussion is hard to follow. The authors should specifically discuss their results in comparison with the previous findings
We thank the reviewer for the comment and the Discussion section was changed accordingly.